# Memory-Efficient 3D Denoising Diffusion Models for Medical Image Processing

**Florentin Bieder**[1]                                    FLORENTIN.BIEDER@UNIBAS.CH
**Julia Wolleb**[1]                                          JULIA.WOLLEB@UNIBAS.CH
**Alicia Durrer**[1]                                        ALICIA.DURRER@UNIBAS.CH
**Robin Sandkühler**[1]                          ROBIN.SANDKUEHLER@UNIBAS.CH
**Philippe C. Cattin**[1]                              PHILIPPE.CATTIN@UNIBAS.CH

[1] *Department of Biomedical Engineering, University of Basel*

**Editors:** Accepted for publication at MIDL 2023

## Abstract

Denoising diffusion models have recently achieved state-of-the-art performance in many image-generation tasks. They do, however, require a large amount of computational resources. This limits their application to medical tasks, where we often deal with large 3D volumes, like high-resolution three-dimensional data. In this work, we present a number of different ways to reduce the resource consumption for 3D diffusion models and apply them to a dataset of 3D images. The main contribution of this paper is the memory-efficient patch-based diffusion model *PatchDDM*, which can be applied to the total volume during inference while the training is performed only on patches. While the proposed diffusion model can be applied to any image generation task, we evaluate the method on the tumor segmentation task of the BraTS2020 dataset and demonstrate that we can generate meaningful three-dimensional segmentations.

**Keywords:** diffusion models, three-dimensional, supervised segmentation

## 1. Introduction

Denoising diffusion models (Ho et al., 2020; Nichol and Dhariwal, 2021) have lately shown an impressive performance in image generation and experienced increasing popularity in medical image analysis (Kazerouni et al., 2022). However, the processing of large three-dimensional (3D) volumes, which often is required in medical applications, is still a challenge. Limitations related to the computational resources only allow the processing of small 3D volumes, which impedes the processing of high-resolution magnetic resonance (MR) or computer tomography (CT) scans.

**Contribution**   In this work, we introduce architectural changes to the state-of-the-art diffusion model implementation (Nichol and Dhariwal, 2021), enabling to train on large 3D volumes with commonly available GPUs. We adapt the U-Net-like architecture to improve the speed and memory efficiency. Furthermore, we propose a novel method illustrated in Figure 1. With this method, the diffusion model is trained only on coordinate-encoded patches of the input volume, which reduces the memory consumption and speeds up the training process. During sampling, the proposed method allows processing large volumes in their full resolution without needing to split them up into patches. To evaluate our method, we perform diffusion model based image segmentation (Wolleb et al., 2022b) that has previously

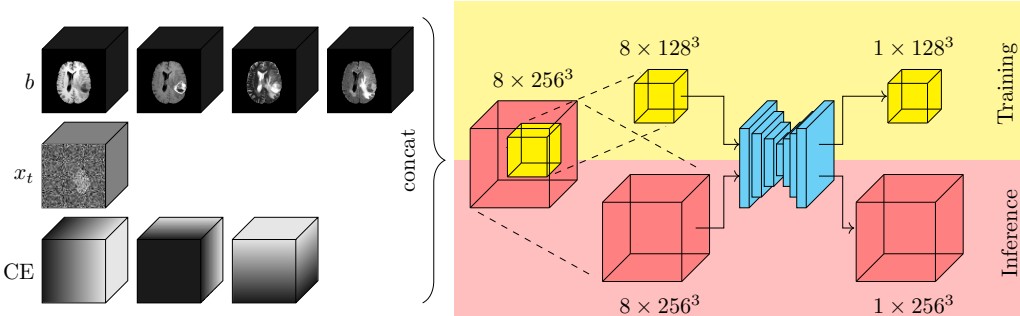

Figure 1: Overview of our proposed method *PatchDDM*. The diffusion model is optimized in memory efficiency and speed by training only on coordinate-encoded patches. The input consists of noised $x_t$, the volumes $b$ that are to be segmented and which are provided as a condition for the segmentation, as well as a coordinate encoding $CE$ for the patches. During sampling, the whole 3D volume can be processed at once.

been proposed for 2D segmentation on the BraTS2020 dataset (Menze et al., 2014; Bakas et al., 2017, 2018). The code is available at github.com/florentinbieder/PatchDDM-3D.

**Related Work**   Denoising diffusion models have seen a quick adoption in research, replacing the more traditional generative models in many tasks such as such as unconditional and conditional image generation (Ho et al., 2020; Song et al., 2021; Nichol and Dhariwal, 2021), text-to-image translation (Nichol et al., 2021; Saharia et al., 2022b; Ramesh et al., 2021; Kim et al., 2022) and inpainting (Ramesh et al., 2021; Nichol et al., 2021). Diffusion models have also been used for various applications in the medical field, for instance, for anomaly detection (Wolleb et al., 2022a), synthetic image generation (Dorjsembe et al., 2022; Peng et al., 2022) and segmentation (Guo et al., 2022; Wu et al., 2022; Wolleb et al., 2022b). Medical images, however, often are 3D volumes, such as MR- or CT-scans. These volumes create challenges regarding the memory consumption of processing methods. Consequently, many of the current methods are limited to two-dimensional (2D) slices only (Wu et al., 2022; Wolleb et al., 2022b; Guo et al., 2022) or to 3D volumes restricted to a limited resolution of at most $128 \times 128 \times 128$ (Khader et al., 2022; Peng et al., 2022; Dorjsembe et al., 2022). To the best of our knowledge, we are the first to tackle the challenge of applying denoising diffusion models to large 3D volumes.

## 2. Method

We explore how denoising diffusion implicit models (DDIMs) presented in Section 2.1 can be improved regarding memory efficiency and time consumption. The required architectural changes are presented in Section 2.2. We present the training and sampling scheme of our method *PatchDDM* in Section 2.3. For evaluation, we use the segmentation approach using denoising diffusion models presented in Section 2.4.

## 2.1. Denoising Diffusion Models

In the following, we will use the notation introduced by (Ho et al., 2020). Denoising diffusion models rely on an iterative noising and denoising process. The forward noising process $q$ is given by

$$q(x_t \mid x_{t-1}) = \mathcal{N}(x_t; \sqrt{1 - \beta_t} x_{t-1}, \beta_t I) \tag{1}$$

where $\beta_t$ is a predefined sequence of variances. We can directly compute $x_t$ from a given $x_0$ with

$$q(x_t \mid x_0) := \mathcal{N}(x_t; \sqrt{\overline{\alpha_t}} x_0, (1 - \overline{\alpha_t}) I) \tag{2}$$

with $\alpha_t := 1 - \beta_t$ and $\overline{\alpha_t} := \prod_{s=1}^{t} \alpha_s$. This corresponds to degrading the input image by adding Gaussian noise. For image generation tasks we are interested in the reverse process $p_\vartheta$.

$$p_{\vartheta,t}(x_{t-1} \mid x_t) = \mathcal{N}(x_{t-1}; \mu_{\vartheta,t}(x_t), \Sigma_{\vartheta,t}(x_t)) \tag{3}$$

Both $\mu_{\vartheta,t}$ and $\Sigma_{\vartheta,t}$ can be estimated by a U-Net-based network $\varepsilon_{\vartheta,t}$ with parameters $\vartheta$. The loss used to train the network $\varepsilon_{\vartheta,t}$ can be written as

$$\|\varepsilon - \varepsilon_{\vartheta,t}(x_t, t)\|^2 = \|\varepsilon - \varepsilon_{\vartheta,t}(\sqrt{\overline{\alpha_t}} x_0 + \sqrt{1 - \overline{\alpha_t}} \varepsilon, t)\|^2, \quad \text{with } \varepsilon \sim \mathcal{N}(0, I). \tag{4}$$

Using the DDIM (Song et al., 2021) sampling scheme, we can define

$$x_{t-1} = \sqrt{\overline{\alpha_{t-1}}} \left( \frac{x_t - \sqrt{1 - \overline{\alpha_t}} \varepsilon_{\vartheta,t}(x_t)}{\sqrt{\overline{\alpha_t}}} \right) + \sqrt{1 - \overline{\alpha_{t-1}}} \varepsilon_{\vartheta,t}(x_t), \tag{5}$$

where $\varepsilon_\vartheta(x_t)$ is the output of the network. This sampling scheme has the advantage that the denoising process is deterministic and we do not need to sample random vectors in every step. Thus, the only source of stochasticity during inference originates from the random initial sample $x_T$ which is sampled from $\mathcal{N}(0, I)$. During inference, a sequence of images $x_i$ for $i = T, T-1, \ldots, 0$ of decreasing noise level is being generated, the initial $x_T$ is sampled from a standard normal distribution $\mathcal{N}(0, I)$.

## 2.2. Architecture

We adapt the 2D-U-Net-based network architecture proposed by (Ho et al., 2020; Nichol and Dhariwal, 2021) and used by (Nichol et al., 2021; Bansal et al., 2022; Wu et al., 2022; Song et al., 2021) for the application on 3D data. The previously proposed architecture features two or three residual convolutional blocks at each down- and upsampling step. Furthermore, it uses attention blocks at multiple resolutions as well as in the bottleneck. (Saharia et al., 2022a) determined that adding global self attention can slightly improve the quality of the generated images as compared to an increase in convolutional blocks. For 3D data, the attention blocks use disproportionally more memory, which made it infeasible to use on current hardware, which is why we removed them completely. The second fundamental change we implemented was the use of additive skip connections, as shown in Figure 2. In the previous architecture as well as in the original U-Net implementation (Ronneberger et al., 2015), the skip connection uses concatenation to combine $x_s$ from the encoder with the upsampled tensors $x_u$ from the lower resolution path of the decoder. This implies that

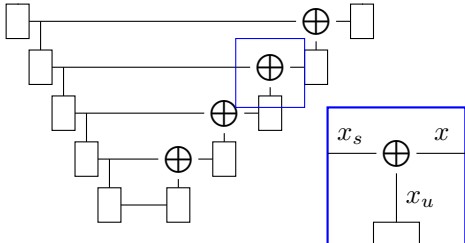

Figure 2: The architecture of the U-Net-like network with averaging skip connections. In the original network as well as in the U-Net the $\bigoplus$ operator is a concatenation $x = (x_s, x_u)$, in our case it is an averaging operator $x = (x_s + x_u)/2$.

the decoder requires significantly more resources than the encoder, especially at the highest resolution levels. To alleviate this issue, we propose to average them as $x = \frac{1}{2}(x_s + x_u)$. Unlike in ResNet (He et al., 2016), where the skip connections are added, we found the averaging to be crucial for avoiding numerical issues like exploding gradients. Intuitively this can be justified by considering $x_s$ and $x_u$ as random variables with $x_u, x_s$ iid. $\mathcal{N}(0, \sigma^2)$. Therefore, $x_u + x_s \sim \mathcal{N}(0, 2\sigma^2)$, that is, with each concatenation the variance doubles, while averaging preserves the variance.

The savings in memory from replacing the concatenation with the averaging allow us to increase the network width, i.e. the number of channels within the whole network, by a factor of 1.61, while preserving the total memory usage. Furthermore, the resulting network architecture allows for training on varying input sizes. This property is crucial for our proposed patch-based method. For all of our experiments, we use the same network configuration.

### 2.3. Patch-based Approach with Coordinate-encoding

To benefit from the lower requirements of computational resources but still to operate on the original resolution, we propose a novel patch-based training method named *PatchDDM* that trains on randomly sampled patches of the input but can afterwards be applied to the full resolution volume during inference. This means for the training we can benefit from using smaller inputs, which means we need fewer computations per iteration as well as less memory. For the inference, however, we can pass the entire input volume at once *without* having to sample patches and reassemble them. This means no boundary artifacts due to the separate padding of the patches within the CNN are introduced, and also the stitching artifacts that that can appear in traditional patch-based approaches are eliminated.

To add information about the position of the patch, we condition the network on the position of the sampled patch. We implemented this by concatenating a grid of Cartesian coordinates to the input. Each coordinate is represented by one channel as a linear gradient ranging from -1 to 1. This is similar to the method proposed in (Liu et al., 2018). They propose to add the coordinates as additional channels before all convolutions.

In our case, we append the coordinates to the whole input just like in (Liu et al., 2018), but then sample a patch, where the coordinates serve as a position encoding for the sampled patch. An overview of this coordinate encoding is given in Figure 1. For the BraTS2020 data, the subject is centered within the volume. We use a patch sampling strategy, assigning a higher probability to the center of the volume, as shown in Figure 6 in the Appendix A.

**Baseline methods** For our ablation study, we use two baselines with the same network as our proposed approach, but without patch-based training. Furthermore, we also performed an experiment with our patch based approach but without the proposed coordinate encoding. The training did not converge and did not produce any usable results. Therefore, we will not report any metrics from this experiment. The two baseline methods are the following:

- Training on full resolution (*FullRes*): We implemented a distributed version of the proposed architecture that splits the task to two GPUs if necessary. This allows for training directly on full resolution ($256^3$) data, given that the expensive specialized GPU hardware is available.

- Training on half resolution (*HalfRes*): A straightforward way to reduce the requirements in terms of computational resources is training the model on downsampled data. In our experiments, we downsampled the input image before passing it to the network, but then upsampled the output of the network again to evaluate the performance on the full size. For three spatial dimensions (i.e. 3D) this means that reducing the input size from $256^3$ to $128^3$ results in a reduction of a factor of 8 in terms of memory and computation time, allowing this model to be run on widely available GPUs.

### 2.4. Denoising Diffusion Models with Ensembling for Segmentation

In order to generate the segmentation of an input image $b$, we need to condition the generation of the segmentation mask $x_0$ on that given image $b$. We will follow the method proposed by (Wolleb et al., 2022b), where the input images $b$ are being concatenated to every $x_t$ as a condition. It was shown that ensembling several predicted segmentation masks per input image increases the segmentation performance (Amit et al., 2021; Wolleb et al., 2022b). An overview of this segmentation approach is given in Figure 3. An advantage of the denoising diffusion based segmentation approach is the implicit ensembling we get when using different samples $x_T$ from the noise distribution $\mathcal{N}(0, I)$, which can be used to increase the performance and estimate the uncertainty. To evaluate the performance of our proposed method *PatchDDM* described in Section 2.3 and the two baseline methods *FullRes* and *HalfRes*, we apply our method to a segmentation task as proposed in (Wolleb et al., 2022b). Therefore, we train our diffusion model to generate semantic segmentation masks.

## 3. Experiments

**Dataset** For our experiments, we used the BraTS2020 dataset (Menze et al., 2014; Bakas et al., 2017, 2018). It contains 369 head MR-scans, each including four sequences (T1, T1ce, T2, FLAIR) with a resolution of $1 \times 1 \times 1$ mm$^3$, resulting in a total scan size of

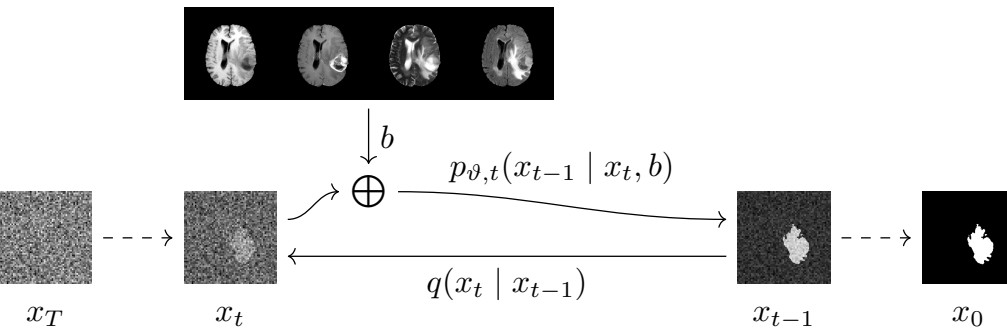

Figure 3: The ground truth segmentation $x_0$ is degraded by the noising process $q$. We train a network to perform the denoising process $p_\vartheta$, that is, given some noised image $x_t$, we train it to denoise it with the MR-sequences $b$ as a condition.

$240 \times 240 \times 155$, which we padded to a size of $256 \times 256 \times 256$. The background voxels were set to zero and the range between the first and 99th percentile was normalized to $[0, 1]$. We used an $80\%/10\%/10\%$ split for training, validation and testing. The label masks consist of three classes, namely the Gadolinium-enhancing tumor, the peritumoral edema, and the necrotic and non-enhancing tumor core. For the binary segmentation experiments, all three classes were merged into one.

**Training Details** We performed our experiments on NVIDIA A100 GPUs with $40GB$ of memory each. To directly train on the full resolution $256^3$ images, we distribute the model over 2 GPUs. The methods *HalfRes* and *PatchDDM* were trained on one GPU only. The optimizer we used was AdamW (Loshchilov and Hutter, 2017) with the default parameters. We chose the learning rate lr $= 10^{-5}$ by optimizing the average Dice coefficient on the validation set after 150k optimization steps over a range of values between $10^{-6}$ and $10^{-3}$. We trained the models for the same amount of time for all experiments (420h). For the evaluation, we selected the best-performing models based on the average Dice score on the validation set based on a single evaluation, i.e., without ensembling. For the denoising process, we set the number of steps to $T = 1000$ and use the affine variance schedule proposed in (Ho et al., 2020) with $\beta_1 = 0.02, \beta_T = 10^{-4}$.

**Accelerated Sampling** By default, we need $T = 1000$ denoising steps for the inference. As shown in (Song et al., 2021), we can interpret the the DDIM denoising step (5) as the Euler discretization of an ordinary differential equation (ODE). This insight motivates the use of larger step sizes with respect to $t$ during inference, which allows for accelerated sampling. The drawback is that the output quality deteriorates with fewer samples. We investigate how we can trade off fewer sampling steps (larger step sizes) and ensembling (more samples).

## 4. Results

In the following, we will assess the performance of our proposed model and compare it to the two baseline approaches. For each model, we computed the average Dice score on the validation set and used this to choose the best-performing checkpoint. We provide some qualitative outputs in Figure 7 in the Appendix B. To assess the training progress, we display the Dice score as well as the HD95 (Hausdorff distance, 95th percentile) of *PatchDDM* over the course of the training in Figure 8 in the Appendix C. The metrics of the best-performing checkpoint with respect to the Dice score when using a single evaluation (no ensembling) is reported in Table 2 in Appendix D along with the score of the state of the art nnU-Net (Isensee et al., 2021).

### 4.1. Segmentation Ensembling

To evaluate the impact of ensembling, we compute the Dice- and HD95-score of the three methods (*PatchDDM*, *FullRes*, *HalfRes*) with respect to ensemble size, see Figure 4. Both scores significantly improve using ensembles for our proposed *PatchDDM* and the *FullRes* method. In Table 3 in Appendix E the metrics for different ensemble sizes are provided. The curves show that ensembling can further improve the performance and get very close to the best performing ensembles with an ensemble size of as small as five to nine.

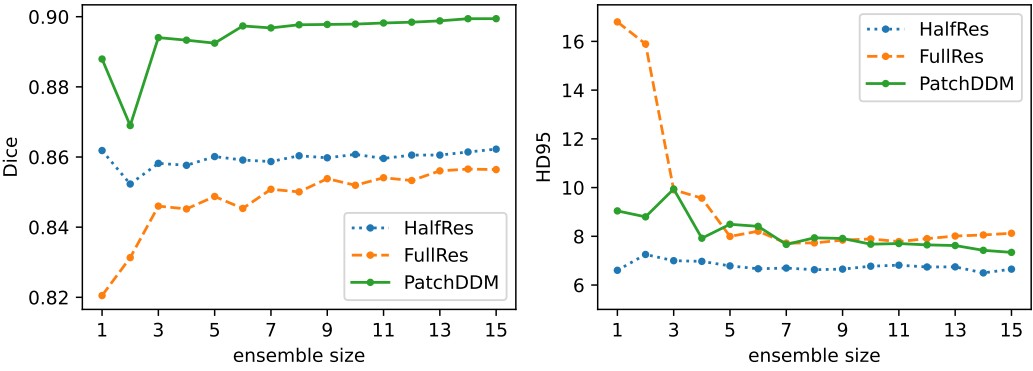

Figure 4: The evaluation metrics on the test set as a function of the ensemble size.

### 4.2. Computational Resources & Time Requirements

We report the memory consumption and the time required for one model evaluation for all comparing methods. As displayed in Table 1, the training of *FullRes* needs close to 80GB of memory. This requires at the time of writing still highly expensive hardware. The other baseline *HalfRes* as well as our proposed *PatchDDM* method both need less than 12GB for training and can therefore be trained on much cheaper and widely available hardware. The reduced resolution also results in a reduction in the number of computations, and therefore a larger number of optimization steps that can be performed in a given time interval. A

drawback of our proposed method is the increased memory consumption and reduced speed during inference, both of which are comparable to the *FullRes* model.

Table 1: Memory consumption in GB and time in seconds for one network evaluation. The memory requirements for the distributed run also include a small amount of overhead, as some arrays are duplicated on both GPUs.

| Method | Memory | | Time | |
| --- | --- | --- | --- | --- |
| | Training | Inference | Training | Inference |
| *FullRes* | 78.5 | 25.7 | 2.12 | 1.01 |
| *HalfRes* | 10.5 | 4.90 | 0.351 | 0.124 |
| *PatchDDM* | 10.6 | 24.0 | 0.340 | 1.02 |

### 4.3. Ensembling and Accelerated Sampling

Figure 5 shows the trade-off between the ensemble size and the number of sampling steps. With as little as 20 sampling steps (i.e. a step size of 50), the performance is already close to the results obtained with $T = 1000$ steps, implying a speedup of a factor of 50. But even with fewer step sizes, we can trade the number of steps for a greater ensemble size to achieve a similar performance. Consequently, for a fixed budget of network evaluations (i.e. steps), we can profit from using ensembling with accelerated sampling.

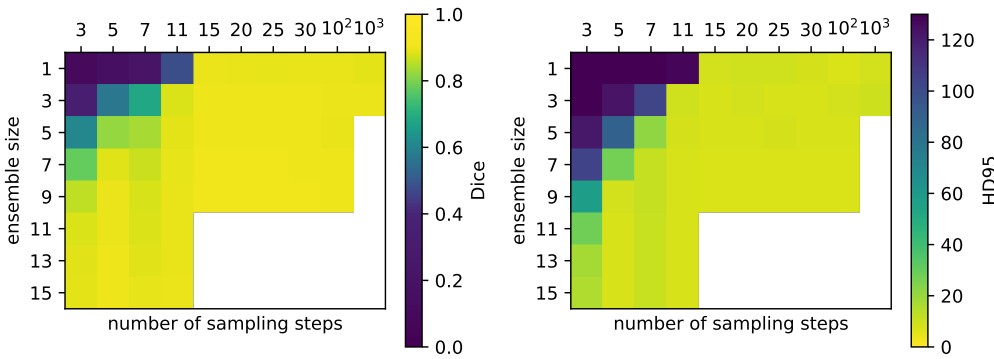

Figure 5: The average Dice score and HD95 metric on the test set as a function of the number of sampling steps and the ensemble size. The white sections indicate that we did not evaluate that combination.

## 5. Discussion

We propose *PatchDDM*, a novel patch-based diffusion model architecture that allows the training of diffusion models on high-resolution 3D datasets. This enables denoising diffusion models to be used for image analysis and -processing tasks in medicine on commonly available hardware. We could demonstrate the effectiveness by applying it to a recently developed segmentation framework for medical images. In the future, we would like to investigate the performance of our proposed approach for tasks involving image generation. Furthermore, we will investigate the role of the patch size used and whether it can be made smaller for processing even higher resolution volumes. In order to preserve high quality, (Karras et al., 2022) proposed using higher-order ODE solvers, like the Heun method, when choosing larger step sizes. This might further reduce the number of iterations needed. Finally, it would be interesting to investigate an extension of this segmentation framework that includes multiple classes.

## Acknowledgments

We are grateful for the support of the Novartis FreeNovation initiative and the Uniscientia Foundation (project #147-2018). We would also like to thank the NVIDIA Corporation for donating a GPU that was used for our experiments.

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

## Appendix A. Sampling Distribution

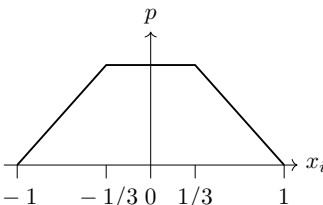

Figure 6: The sampling distribution was chosen empirically to favor the central patches. The distribution $p$ is defined over the normalized coordinates of admissible patches (normalized to $[-1, 1]$) and can be interpreted as the probability density function the sum $X + Y$ of two random variables $X \sim U[-1/3, 1/3], Y \sim U[-2/3, 2/3]$.

## Appendix B. Qualitative Results

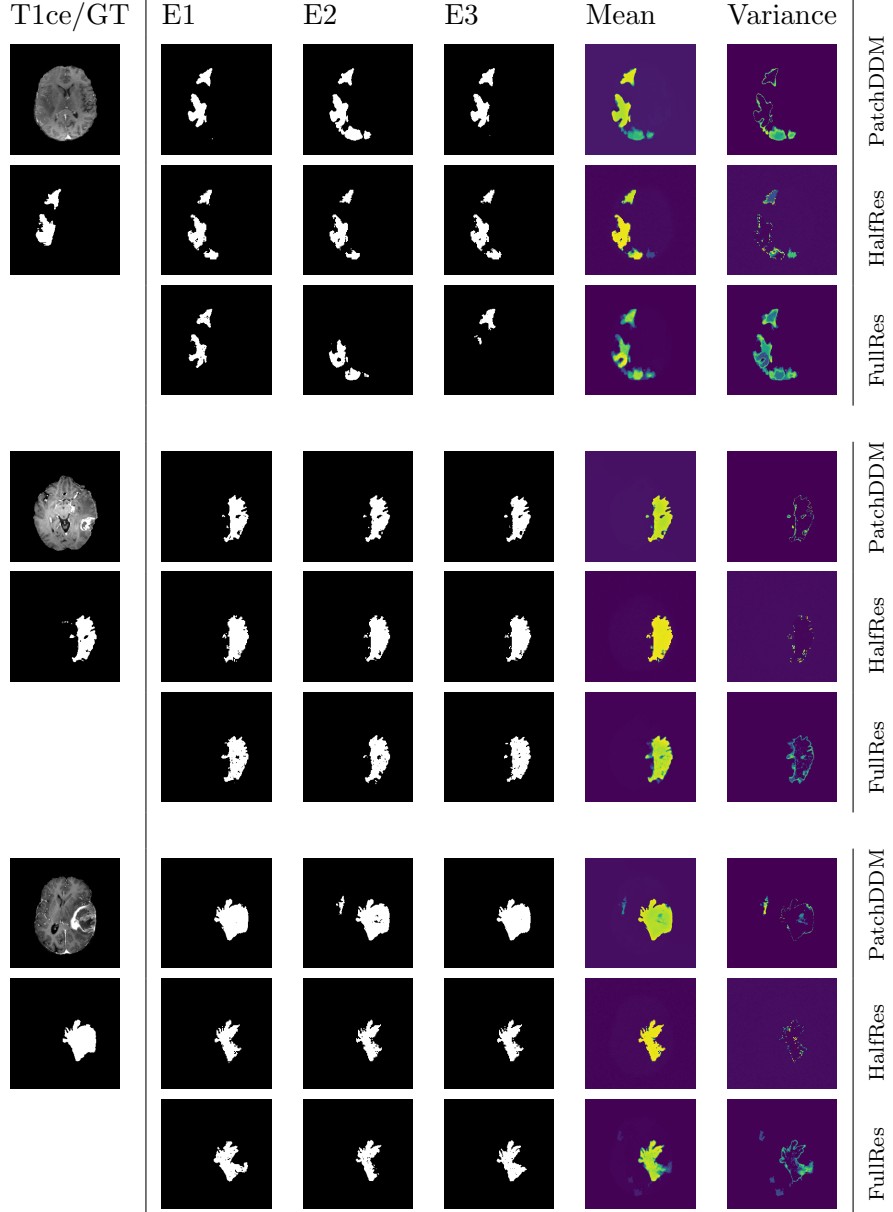

Figure 7: We display an axial slice of three volumes. The first column shows T1ce-sequence and the ground truth segmentation. Then we display three outputs E1-E3 of the ensemble for each of the models and finally the mean- and (normalized) variance map across the ensemble of size 15.

## Appendix C. Training Progress

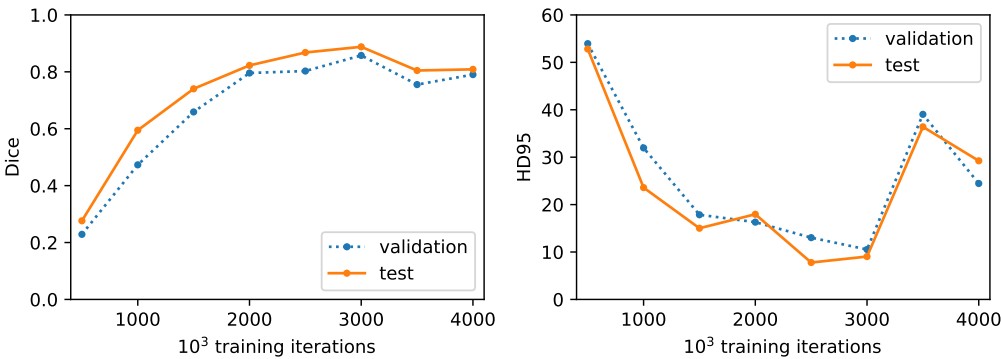

Figure 8: Performance of our method *PatchDDM* on the validation- and test set over the course of the training. The *x*-axis indicates the number of training iterations as a multiple of 1000.

## Appendix D. Single Evaluation Scores

Table 2: Segmentation scores of our methods and nnU-Net on different metrics on our test set based on a single evaluation.

| Method | Dice | HD95 |
|---|---|---|
| *FullRes* | $0.82 \pm 0.12$ | $16.80 \pm 18.96$ |
| *HalfRes* | $0.86 \pm 0.09$ | $6.61 \pm 9.37$ |
| *PatchDDM* | $0.88 \pm 0.07$ | $9.04 \pm 8.75$ |
| nnU-Net | $0.96 \pm 0.02$ | $1.24 \pm 0.48$ |

## Appendix E. Ensembling Scores

Table 3: Segmentation scores of the three methods with various ensemble sizes.

| Method | Ensemble size | 1 | 3 | 5 | 7 | 15 |
|---|---|---|---|---|---|---|
| *FullRes* | Dice | 0.821 | 0.846 | 0.849 | 0.851 | 0.856 |
| | HD95 | 16.80 | 9.91 | 8.00 | 7.72 | 8.13 |
| *HalfRes* | Dice | 0.862 | 0.858 | 0.860 | 0.859 | 0.862 |
| | HD95 | 6.61 | 7.00 | 6.79 | 6.70 | 6.65 |
| *PatchDDM* | Dice | 0.888 | 0.894 | 0.892 | 0.897 | 0.899 |
| | HD95 | 9.04 | 9.94 | 8.49 | 7.67 | 7.34 |

