# OpenReview forum: "Memory-Efficient 3D Denoising Diffusion Models for Medical Image Processing"
_MIDL.io/2023/Conference — MIDL 2023 Poster_

### Official Review · Reviewer_yXo9 · 2023-02-03

**Confidence:** 4
**Preliminary Rating:** 2

**Summary:**

The authors propose a patch based diffusion model for 3D medical image synthesis. They split a 3D volume into small patches to train a diffusion model and use the entire volume for inference to save computational resources. In the experiments, the BraTS 2020 dataset is used for the brain tumor segmentation task. They augment a segmentation model by synthesizing image-mask pairs using a conditional diffusion model. The proposed method improved segmentation performance and reduced computing consumption.

**Strengths:**

- The study is well-motivated that diffusion models are expensive to train and inference especially for 3D medical images;
- The proposed method is straightforward;
- The paper basically conveys the main points and overall well-written;


**Weaknesses:**

- There’s no image synthesis quality comparison between the two baseline methods and the proposed model;
- In Figure 4, no baseline segmentation model that without synthesized image-mask pair augmentation for comparison;
- It doesn't make sense that the full-resolution synthesis strategy underperforms the patchDDM in dice score;
- From Table 1 & Figure 4, half-resolution is the optimal choice in terms of both computational efficiency and segmentation performance compared to the proposed patch-based one;


**Deanonymize Review:**

no

**Paper Type:**

methodological development

**Questions To Address In The Rebuttal:**

- Evaluate image synthesis quality in SSIM, PSNR etc;
- Add necessary segmentation baseline;
- Explain why full-res approach doesn’t perform well;
- Justify on why the proposed method is a better option than half-res one;

---

### Official Review · Reviewer_YB5N · 2023-02-05

**Confidence:** 4
**Preliminary Rating:** 4

**Summary:**

The authors explore ways to make Denoising Diffusion models more efficient to the point they can run on more accessible hardware.

The authors main improvements are:
-  A 3D unet, based on an existing 2D UNet architecture, with several efficiency improvements: the removal of an attention block and the replacement of a concatenation step by an averaging step. These changes reduce the memory consumption.
- They augment the input image with a positional encoding. This allows them to use patches during training instead of using a full volume, and still have spatial information.

The method is applied to a segmentation task in a Diffusion setting, and the patch-based learning is compared to non-patch full volume approaches. The PatchDDM compares well and even outperforms full image DDM in some cases on this particular task.

**Strengths:**

The approaches made in the paper are easy to understand and seem easy to apply.
Any approaches that make the powerful but expensive Diffusion models more efficient are of high value to the ML and medical community.


**Weaknesses:**

The main weakness is that the method is only tested on a segmentation task; not the hardest or most common task of diffusion models.
The authors do seem to acknowledge, in their future work section, that image generation is their next task to try out.

In a segmentation task, there is a clear relation between the input pixel and the desired output.
This makes it hard to test the hypothesis that the positional encoding is enough to put PatchDDM on par with Full image DDM for more complex tasks.
Is the positional encoding even needed for example, or does patchDDM also have good results without it?

The results are also not put in the context of other methods on the Brats segmentation task.
The results look competitive, some quick searching shows the winning result in 2020 had 88.9 Dice where this method seems to have 89.9? Also, I have to read these values from the graph since the numbers are not actually written in the paper.


**Deanonymize Review:**

no

**Detailed Comments:**

Some details of the implementation are not so clear. E.g. how is the coordinate embedding done? The paper refers to another paper in which it might be explained, but I would rather see a quick explanation of this detail, than the review of the Diffusion equations in 2.1. Ideally both are present in the paper.

Also how exactly are patches samples and at what resolution?

Also the ensembling explanation is kind of hidden at the end of 2.4 which took me some time to find. This could be explained more clearly.

Also, the segmentation results are not compared to existing methods. How does this work compare to whatever is the best scoring method on this task?

Additionally, some more discussion on the results would be interesting. PatchDMM gets much higher Dice than even the full res model. And the half resolution model for some reason scores the best in the Hausdorf distance. Some kind of argumentation would be interesting here.

**Paper Type:**

methodological development

**Questions To Address In The Rebuttal:**

I think the argument for reduction of memory usage is clear, but I wonder if the authors can strengthen their argument why this segmentation application is a convincing testament for the method working well. E.g. could one say that these results are somehow state of the art, and with the more efficient method classification via Diffusion models is closer to becoming applicable in real settings?

---

### Official Review · Reviewer_8hvs · 2023-02-06

**Confidence:** 3
**Preliminary Rating:** 4
**Recommendation:** Poster

**Summary:**

The paper focuses on the optimization of diffusion models for memory-efficient processing of 3D medical images. The motivation of the paper is very clear and the paper is mostly methodological. It focuses on main improvements of existing diffusion models in order to reduce memory usage, while accelerating the training. The main idea relies on dividing the large 3D volumes into patches. This allows for much more efficient training times but the inference times are not as optimal.

**Strengths:**

The main strength of the paper is the optimization of the denoising diffusion models to be able to be applied with inexpensive GPUs on 3D images, which is one of the main challenges in the medical imaging community. Moreover, they show that using this small improvements, they perform better than the original denoising diffusion models on the segmentation task of the BraTS2020.

**Weaknesses:**

It would be nice if the methods were compared on more than 1 datasets. A comparison of qualitative results with full, half and patch DDM is also missing. Even though the dice of the presented method is higher, it would be nice to see how does it qualitatively fail. In general this would help on the validation of the method.
The memory usage and time is very nicely optimized during training but not so much during inference. It would be great to find a way to improve that. Moreover, it would be great if the authors would also include a further explanation on why the boundary artifacts that appear in traditional patch-based approaches do not appear at inference time.

**Deanonymize Review:**

no

**Paper Type:**

methodological development

**Questions To Address In The Rebuttal:**

1. How does the inference work? How is the relation between the learned patches and the inferred total 3D volume?
2. What could be a possible improvement of the paper in order to achieve similar inference evaluation memory and computational times as the half resolution solution?

---

### Meta-Review · Area_Chair_uqKZ · 2023-02-22

**Recommendation:** Accept (Poster)
**Confidence:** 5

**Metareview:**

Denoising Diffusion Models for Memory-efficient Processing of 3D Medical Images

The consensus is a on weak acceptance, ranging from 2 acceptances and 1 mild rejection. The contribution is on improving the diffusion model by using patches to reduce the overal memory usage. The authors have addressed the main concern on their internal mask synthesis. For these reasons and given the global ranking of submissions, the recommendation is leaning towards acceptance.

Recommendation towards Acceptance.